# Biodegradation Studies of Polyhydroxybutyrate and Polyhydroxybutyrate-*co*-Polyhydroxyvalerate Films in Soil

**DOI:** 10.3390/ijms24087638

**Published:** 2023-04-21

**Authors:** Jihyeon Kim, Nevin S. Gupta, Lindsey B. Bezek, Jacqueline Linn, Karteek K. Bejagam, Shounak Banerjee, Joseph H. Dumont, Sang Yong Nam, Hyun Woo Kang, Chi Hoon Park, Ghanshyam Pilania, Carl N. Iverson, Babetta L. Marrone, Kwan-Soo Lee

**Affiliations:** 1Chemistry Division, Los Alamos National Laboratory, Los Alamos, NM 87545, USA; jhkim@lanl.gov (J.K.); nevin5@comcast.net (N.S.G.); lbezek@lanl.gov (L.B.B.); jacil@lanl.gov (J.L.); joseph.dumont@lanl.gov (J.H.D.); iverson@lanl.gov (C.N.I.); 2Department of Materials Engineering and Convergence Technology, Gyeongsang National University, Jinju 52828, Republic of Korea; walden@gnu.ac.kr; 3Materials Science and Technology Division, Los Alamos National Laboratory, Los Alamos, NM 87545, USA; karthik3327@gmail.com (K.K.B.); gpilania@lanl.gov (G.P.); 4Bioscience Division, Los Alamos National Laboratory, Los Alamos, NM 87545, USA; baners4@lanl.gov (S.B.); blm@lanl.gov (B.L.M.); 5Department of Energy Engineering, Future Convergence Technology Research Institute, Gyeongsang National University, Jinju 52725, Republic of Korea; hskang12@gnu.ac.kr (H.W.K.); chp@gnu.ac.kr (C.H.P.); 6General Electric Global Research Center, Niskayuna, NY 12309, USA

**Keywords:** polyhydroxybutyrate, biodegradable polymers, polymer degradation, green chemistry, density functional theory

## Abstract

Due to increased environmental pressures, significant research has focused on finding suitable biodegradable plastics to replace ubiquitous petrochemical-derived polymers. Polyhydroxyalkanoates (PHAs) are a class of polymers that can be synthesized by microorganisms and are biodegradable, making them suitable candidates. The present study looks at the degradation properties of two PHA polymers: polyhydroxybutyrate (PHB) and polyhydroxybutyrate-*co*-polyhydroxyvalerate (PHBV; 8 wt.% valerate), in two different soil conditions: soil fully saturated with water (100% relative humidity, RH) and soil with 40% RH. The degradation was evaluated by observing the changes in appearance, chemical signatures, mechanical properties, and molecular weight of samples. Both PHB and PHBV were degraded completely after two weeks in 100% RH soil conditions and showed significant reductions in mechanical properties after just three days. The samples in 40% RH soil, however, showed minimal changes in mechanical properties, melting temperatures/crystallinity, and molecular weight over six weeks. By observing the degradation behavior for different soil conditions, these results can pave the way for identifying situations where the current use of plastics can be replaced with biodegradable alternatives.

## 1. Introduction

Plastic waste has become ubiquitous in our society and is often a waste product originating from industrial development. According to 2022 OECD reports, plastic pollution in the environment is expected to significantly worsen in the coming decades. Specifically, the amount of plastic leakage into the environment is anticipated to double to 44 million tons a year, while the buildup of plastic waste in waterways is projected to increase by more than threefold, from 353 million tons in 2019 to over 1000 million tons by 2060 [1]. Most pollution comes from larger debris known as microplastics, but the leakage of these microplastics (synthetic polymers less than 5 mm in diameter) from items like industrial plastic pellets and textiles is also a serious concern. Despite efforts to reduce the proportion of mismanaged plastics, a significant proportion (at least 14 million tons per year) enters the ocean and pollutes the environment [1,2]. Forty percent of plastic waste is produced from packaging materials, which are usually single-use plastics. Packaging material is mainly composed of polypropylene (PP), high density polyethylene (HDPE), low density polyethylene (LDPE), and polyethylene terephthalate (PET) [3]. These polymers are produced from petrochemical derivatives, are not biodegradable, and can persist for centuries, putting significant pressure on the environment [4,5]. Moreover, even though recycling protocols are well established for many plastics, these plastics are recycled at a rate of less than 9% globally [6,7,8]. A total of 88% of the sea surface is polluted by plastic waste. In addition, 56% of sea surface planktonic samples contain microplastic particles [9,10,11,12]. Ingested microplastics have a multiplying effect since they can be transferred up the food chain, which has the potential to harm humans [9,11,12,13]. Therefore, intensive research has recently focused on developing alternatives to biomass-based materials, specifically biodegradable polymers and polymers produced by microorganisms, [14,15,16,17] which have lower carbon footprints for production and less environmental impact when discarded. One of the challenges for such material development is attaining durable material properties and biodegradation performance. Balancing the material and processing capabilities of new bioplastics with biodegradability is a critical need of the sustainable plastic industry in the future.

Polyhydroxyalkanoates (PHAs), one of the representative natural polyester-based biodegradable polymers, are a family of thermoplastic polymers that are produced by a number of bacteria including cyanobacteria. Furthermore, the PHA family is considered biodegradable, non-toxic, eco-friendly, and can be produced from renewable resources [18]. As effective alternatives, PHAs, such as polyhydroxybutyrate (PHB, or P3HB), poly-3-hydroxyvalerate (PHV), poly(hydroxybutyrate-*co*-hydroxyvalerate) (PHBV), poly-4-hydroxybutyrate (P4HB), poly(3-hydroxyoctanoate) (PHO), poly(3-hydroxynonanoate) (PHN), 3-hydroxyhexanoate (HHx), 3-hydroxyheptanoate (HH), and 3-hydroxydecanoate (HD), are promising candidates [19,20]. Table 1 shows the properties of various PHAs. PHAs can have varying molecular weights from 50,000 to over 1,000,000 g/mol and can be broken down by microorganisms. These microorganisms can produce CO_2_ and H_2_O under aerobic conditions, and CO_2_, H_2_O, and CH_4_ under anaerobic conditions when ingested [14,18,21]. Since microorganisms use carbon from their environment to synthesize the polymers, a net-zero carbon footprint is created from synthesis to degradation. This creates a sustainable cycle for the production and degradation of PHAs.

In particular, PHB has gained much attention as a replacement for non-biodegradable commercial polymers as it has similar mechanical strength properties to PE and PP [18,19]. However, as a homopolymer, PHB has a few limitations: it is rigid, brittle, and has low elongation properties due to high crystallinity (Table 1). Furthermore, PHB’s melting temperature is close to its degradation temperature, making it difficult to process industrially [22,23]. The addition of hydroxyvalerate units (HV) generally reduces the crystallinity; improving the mechanical properties, lowering the melting temperature, and making it easier to process and manufacture [24]. PHB is completely biodegradable under various natural active environments such as soil, industrial composting, and seawater by an/aerobic sludge containing several microorganisms [25,26,27]. Likewise, other biopolymers can be degraded by enzymatic activity or microbes, including polylactic acid (PLA), poly(butylene adipate-*co*-terephthalate) (PBAT), and polycaprolactone (PCL). Previously, several studies confirmed the degree of biodegradation of these polymers under soil conditions [28,29,30,31,32]. The studies showed that the biodegradation process is affected by various factors, including polymer structure, morphology, chemical treatment, and molecular weight. Specifically, polymers with a structure containing hydrolyzable linkages and/or a morphology containing amorphous regions are relatively more susceptible to degradation [32,33]. Although the biodegradation of PHB has been broadly investigated, studies have typically focused on screening for PHB-degrading enzymes/bacteria under controlled conditions, and little attention has been paid to observing their performance in familiar environments such as home composting, plant soil, and lakes [34,35]. Despite the potential that PHB shows as biodegradable plastic, a lack of understanding of its degradation in these environments hinders the potential for substituting it for commercial/conventional plastic. In order for these biodegradable polymers to be commercially viable, it is essential to understand their physical and chemical properties, their degradation mechanisms, and how their properties change under various degraded conditions.

**Table 1 ijms-24-07638-t001:** Properties of various PHA-derived polymers.

PHAs	Crystallinity (%)	Melting Point (°C)	Tensile Strength (MPa)	Elongation at Break (%)	Ref.
PHB	60–80	170–180	35–50	3–5	[21,23]
PHBV	39–69	102–157	22–36	8–10	[36]
P4HB	57	60	50	1000	[37,38]
PHO	30	62	6–10	300–450	[36]
PHN *	60	63	15	1317	[39,40]

* 95 mol% HN.

In this study, the biodegradation behavior of polyhydroxybutyrate (PHB) was studied in comparison with polyhydroxybutyrate-*co*-polyhydroxyvalerate (PHBV, 8 wt.% of valerate in Figure 1) to study the kinetics and mechanism of hydrolytic destruction under aqueous and non-aqueous media using commercial plant soil. Wherever possible, theoretical support based on density functional theory (DFT) based calculations is provided to corroborate our experimental observations. We observed the degradation of the PHB and PHBV in soil saturated with water (100% RH) and in soil kept at 40% RH at 25 °C, over six weeks. The contrast between the degradation profiles for PHB and PHBV makes it possible to compare the general degradation behavior for the most prevalent biodegradable PHA polymers. In addition, a detailed comparison of thermomechanical properties as a function of degradation time is presented for the two polymer chemistries. In the following, we present our findings in greater detail.

## 2. Results and Discussion

### 2.1. Initial Observations

The degradation of the 100% RH samples was starkly different than that of the 40% RH samples. Figure 2 shows the degradation of the PHB and PHBV samples under 100% RH soil and 40% RH soil. In 100% RH conditions, both samples showed slight deterioration with minor holes in appearance after three days. By the seventh day, the samples had begun to break down further, as seen by the holes throughout the sample (Figure 2a,b). In the case of PHB samples, the changes may have been masked by the initial white color of the samples, but it should be noted that similar degradation could have been occurring in them. By the 10th day, both the PHB and PHBV samples showed a dramatic change in their integrity. Only a vague shape of the original sample remained, and the remnants of the samples were in broken fragments. After 14 days (not pictured) only the taped ends of the samples remained. In contrast to the degradation of both PHB and PHBV samples under 100% RH soil conditions, the samples under 40% RH showed much less degradation. Even after six weeks of degradation, both the PHB and PHBV samples retained the same appearance as the pristine samples. This indicates that a key factor in the degradation of PHB and PHBV is the degree of humidity. The microscope images of PHB and PHBV after 7 days under 100% RH, and 6 weeks under 40% RH, are shown in Appendix A, respectively.

As we mention in Section 3.3, the soil used for the biodegradation study was slightly acidic. Figure 3 shows the hydrolysis reaction mechanism under acid-based conditions. The degradation findings we observed aligned with our expectations for such materials in acidic conditions, as shown by Rydx et al. [41]. Generally, the acid-catalyzed degradation of polyesters begins with the protonation of the carbonyl oxygen of the ester group by a hydronium ion (H_3_O^+^), which makes the carbonyl carbon more electrophilic due to the positive charge. Water molecules then attack the carbonyl carbon and a tetrahedral intermediate is generated. After that, the tetrahedral intermediate can decompose into a carboxylic acid and alcohol [41].

### 2.2. Chemical Analysis

FT-IR spectroscopy of the pristine and aged films of PHB and PHBV was used to examine the degradation in soil and understand their degradation mechanism. Figure 4 shows the FT-IR peaks of PHB and PHBV films at 1200–900 cm^−1^ and 3500–2700 cm^−1^ under 100% RH conditions in soil for 10 days. Both films showed clear changes in the magnitude of the peaks and absorbance increased with degradation time. The peaks at 1054–1043 cm^−1^ correspond to the C–O–C asymmetric stretching vibration. The production of primary alcohols from ester can lead to a strong C–O asymmetric stretch between 1025 and 1000 cm^−1^ [43]. At 1021 cm^−1^, there was evidence of the presence of the C–OH group, which was expected from the 3-hydroxybutyric acid in the polymer chain (Figure 4b,d). In addition, the absorbance at 3400–3150 cm^−1^ can be attributed to the OH group produced due to the degradation of the oligomer (Figure 4a,c). This suggests that the bacteria and fungi in the soil decomposed the PHB to 3-hydroxybutyric acid due to oligomer hydrolase and PHB depolymerase, which was ultimately oxidized to acetyl acetate [44,45]. In the case of PHBV, the difference in the peak intensity of the C–OH group and C–O–C group was less than that of PHB. We expected that the degradation rate of PHBV would not be as fast as that of PHB (Figure 4d). No changes were observed in the FT-IR spectra for either PHB or PHBV samples under 40% RH conditions. The full spectrum of both humidity conditions for both materials is provided in Appendix A. The samples all showed characteristic peaks for PHB and PHBV, which are given in Appendix A. 

Figure 5 illustrates the three different oligomers (*n* = 2, 3, and 6) and their corresponding infrared spectra obtained using DFT calculations. The intensity of the C–O asymmetric stretch is approximately 1030–1050 cm^−1^ and the O–H stretching is approximately 3280 cm^−1^ which is consistent with the experiments. Also, the peak intensity increases as the oligomer size decreases from a hexamer (config. a) to a trimer (config. b) to a dimer (config. c). This is qualitatively similar to the increase in FT-IR spectra (Figure 4) suggesting polymer degradation. 

### 2.3. Thermal Properties

Differential scanning calorimetry (DSC) was used to see how the melting point and crystallization properties of the polymers changed over time in soil. Figure 6 shows the melting points and the degree of crystallinity (%) for the PHB and PHBV samples exposed to 100% RH and 40% RH in soil. Changes in melting point enabled an understanding of the physical characteristics of the PHB and PHBV polymer chains. As shown in Figure 6a,b, under 100% RH soil conditions, both the PHB and PHBV samples showed a downward trend in the two melting temperatures. An interesting phenomenon was the double melting peaks in the samples of PHB and PHBV. This splitting of the melting peak indicates a formation of crystallites of two different lamellae thicknesses [46]. Previous investigations showed that well-ordered long polymer chains crystallize easily into thicker lamellae, resulting in higher melting temperatures and degrees of crystallinities, whereas short, poorly-ordered chains have decreased lamellae thickness and reduced melting temperatures [47,48,49]. In addition, crystallinity has an important role in changing the accessibility of polymer chains by microorganisms, resulting in a different morphology of the degraded samples. The amorphous regions could be more accessible to enzymatic attack [50]. Therefore, we believe that the observed lower melting temperatures and degree of crystallinities over time are associated with decreased molecular weight upon hydrolysis.

As shown in the melting temperature and the degree of crystallinity as a function of aging time in Figure 6a,b, under the 100% RH soil conditions, the PHB samples showed a significant decrease to 21% of the degree of crystallinity within three days (the degree of crystallinity of PHB pristine; 70%), which is about a 70% decrease in crystallinity compared to the pristine PHB. After three days, the degree of crystallinity remained relatively constant. The PHBV sample also exhibited a decrease to 38% of the degree of crystallinity over seven days (the degree of crystallinity of pristine PHBV; 42%), which was a decrease in crystallinity of approximately 9% compared to the pristine PHBV. It is especially notable that the PHBV sample showed a gradual decrease in the degree of crystallinity over the course of the experiment, whereas the PHB sample exhibited a large decrease in the short term. A similar trend for the PHB and PHBV samples aged in the 40% RH soil conditions, and especially the crystallinity of the samples, can be seen in Figure 6c,d. Under 40% RH conditions, they showed considerably fewer changes than 100% RH conditions. In fact, the only significant change was seen in the degree of crystallinity of PHB. The PHB samples showed a decrease to 49% in the degree of crystallinity over seven days, which was a decrease in crystallinity of approximately 30% compared to the pristine PHB; a much smaller change than that seen in the samples under 100% RH conditions. Normally, amorphous regions are more susceptible to hydrolysis than crystalline regions [50,51]. The preferential enzymatic attack of the amorphous phase of PHB has been reported [50,52]. However, in this study both the amorphous and crystalline regions were degraded without preference. Furthermore, their enthalpy of melting and consequently the degree of crystallinity also decreased, which may be due to the disruption of the crystalline structure of PHB. Other properties that affect the rate of decomposition of biodegradable materials include the size, shape, frequency of occurrence of crystalline phases, and number of crystallites. In the case of PHBV, we believe that the increased hydrophobicity caused by the hydroxyvalerate (HV) monomer units helped to inhibit water molecules from breaking down the polymer chain [52].

### 2.4. Mechanical Properties

Tensile tests were used to determine how the mechanical properties of PHB and PHBV changed as the samples degraded. Table 2 shows the tensile data for PHB and PHBV samples for 100% RH and 40% RH soil conditions. Under 100% RH soil conditions, the PHB and PHBV samples had less than 1% elongation at break after three days. The ultimate tensile strength also decreased by more than half. After seven days, both samples had voids due to degradation so they could not be measured in the tensile test. It can be seen that both materials in 100% RH soil conditions experienced lower mechanical properties over time. Since elongation and tensile strength are highly dependent on the molecular weight of polymer molecules, a decrease in the mechanical properties would indicate degradation, which is consistent with what was observed in Section 2.5. As the average chain length of the polymers decreases, as suggested in Section 2.3, there are fewer points of entanglement, which leads to reduced elongation and tensile strength. Moreover, shorter polymer chains form fewer Van der Waals interactions and hydrogen bonds, which is consistent with the lower mechanical properties observed. 

Under 40% RH soil conditions, both samples showed no change in the ultimate tensile strength over 43 days. Although we observed a slight decrease in the elongation at break in the samples, there was no considerable change over a month, and both samples exhibited relatively constant mechanical properties over the duration of the experiment. From the tensile results, comparing 40% RH samples to the 100% RH samples, it is clear that the decrease in mechanical properties was much greater in the 100% RH soil conditions. This suggests that the samples degrade very slowly without aqueous media. The stress–strain curves for each sample are shown in Appendix A. 

### 2.5. Molecular Weight Analysis

Size exclusion chromatography (SEC) was used to analyze how the molecular weight of the polymers changed during the degradation experiment over time, which enables the degradation of the polymer to be empirically shown. Figure 7 shows the SEC results for the PHB and PHBV samples for both the 100% RH and 40% RH conditions. As shown in Figure 7, both samples showed a rapid linear decrease in molecular weight. 

These results were expected because most biodegradable polymers are initially broken down into oligomers and monomers, which are subsequently metabolized and degraded. This is because most intact polymers are too large to pass through cellular membranes, so they must first be depolymerized into smaller monomers before they can be absorbed and biodegraded within microbial cells [53]. 

In addition, the decrease in molecular weight for PHB in 40% RH soil appeared to exhibit two different degradation stages. From 7 days to 3 weeks, the samples had a much faster rate of degradation when compared to weeks 3–7. The PHBV samples, on the contrary, showed a gradual, linear decrease in molecular weight over the course of the experiment. As mentioned in Section 2.3, this is mainly due to the hydroxyvalerate (HV) moieties providing hydrophobicity in the polymer chain. The primary mechanism of degradation occurs through a two-step process starting with hydrolysis and followed by the metabolism of microorganisms on the fragmented residues [54,55]. During the first stage of degradation, the high molecular weight polyester chains are quickly hydrolyzed into lower molecular weight oligomers, which are further accelerated by microorganism metabolism and the surrounding environment [54,55]. In particular, the rate of hydrolysis is dependent on the moisture content and temperature, so the polymers degraded much faster under 100% RH soil conditions. PHB products rapidly degrade in both aerobic and anaerobic environments [54,56,57]. As shown in the density functional theory (DFT) calculation for the free energy change in the PHB hydrolysis reaction (Appendix A), the first step is a protonation of carbonyl groups from the hydronium ions. In a water medium, the free energy level of the protonated form is lower than that of the non-protonated form with carbonyl groups. Accordingly, the protonation of the carbonyl groups in polyesters starts spontaneously and the acid-catalyzed hydrolysis can be accelerated under higher RH conditions. Table 3 shows the summary of the changes on M_n_, T_m_, and degree of crystallinity with aging time of PHB and PHBV.

## 3. Materials and Methods

### 3.1. Materials

PHB granules (BU396312) and PHBV films (BV301010) containing 8% HV were purchased from Goodfellow (Coraopolis, PA, USA). The PHBV films were used as received. The PHB samples were cast into thin films. High performance liquid chromatography (HPLC) grade chloroform (Fisher Scientific, Waltham, MA, USA) was used for both making PHB films and SEC experiments. For the degradation study, Miracle-Gro^®^ moisture-controlled potting mix soil was used from Miracle-Gro (Marysville, OH, USA).

### 3.2. PHB Film Preparation

A ~2.5% (*w*/*w*) solution of PHB in chloroform was prepared. The samples were heated to 50 ± 5 °C for 3 h to ensure the polymer did not degrade during the dissolution process; polymer solutions heated too much during dissolution turned a dark brown color indicating degradation (above 100 °C in chloroform, degradation of PHB may occur due to random and chain-end scission). Once the PHB had fully dissolved, the solution was allowed to cool to room temperature. Then the solution was cast over a glass substrate (15 cm × 15 cm) at room temperature. A foil lid with holes was placed over the PHB film in order to slow the rate of chloroform evaporation as this produced consistent, uniform films. The chloroform evaporated for 30 min at room temperature and then the samples were dried in a vacuum for 2–4 h to remove any residual chloroform. The PHBV films had a thickness of 10 ± 2 μm and the prepared PHB films had a thickness of 15 ± 5 μm. Samples were cut into rectangles of 12 mm × 101 mm with an ASTM D882 specimen cutting die.

### 3.3. Soil Degradation

For testing degradation in soil, a glass container was filled with approximately 1 in. of soil. A total of 5 samples for each condition were then placed between two mesh nets, positioned on top of the soil in the container, and covered by another layer of soil. For the 100% RH conditions, the soil was then saturated with deionized water. A cover with a slit cut in it was placed over the samples to allow oxygen and other gases to be exchanged freely during the degradation process. Deionized water was added as necessary to keep the soil saturated and the containers were left at room temperature for up to 14 days. For 40% RH conditions, glass containers including the samples, soil, and a small amount of deionized water were placed in a climate chamber for up to 43 days. The soils were slightly acidic with a measured pH value of 6.6–6.7, even with aging time. This may be due to the very low amount of testing films compared to the amount of soil in each testing batch. These values were measured using a Fisher Scientific accumet AB15 pH meter (Fisher Scientific, Waltham, MA, USA). For these experiments, soil samples were mixed and allowed to settle in ultrapure water (Barnstead, NH, USA). A VHX 6000 digital microscope from KEYENCE (Osaka, Japan) was used to characterize the surface image of the aged films. A microscope was conducted with full ring lighting and the magnifications used were 20× (Appendix A). The experimental procedure of soil degradation is shown in detail in Appendix A.

### 3.4. Fourier Transform Infrared Spectroscopy (FT-IR)

The samples were removed from the soil, washed in distilled water to clear the surface, dried overnight at room temperature in a vacuum oven, and then analyzed using FT-IR. An average of 32 scans were taken using a Nicolet iS50 (Thermo Fisher Scientific, Ogden, UT, USA) producing a resolution of 4 cm^−1^. The analyzed data were normalized to the peak associated with the C–H group of 1456 cm^−1^.

### 3.5. Tensile Testing

Tensile tests were conducted to understand the stress–strain behavior of polymer samples using an Instron 3343 Single Column Testing System (Instron Bluehill®, Norwood, MA, USA). This test was performed following the ASTM D882 standard using a rectangular-shaped specimen (12 mm × 101 mm). To enhance the gripping surface, each sample was affixed with Scotch tape (3M, Saint Paul, MN, USA) of 25 mm in length at both sample ends. The gauge length was 50 mm and tests were run at 1.00 mm/min until break. Each tension data point is an average value of 5 measures of 5 different samples, with an error range of less than 3%.

### 3.6. Differential Scanning Calorimetry

Samples were measured using a DSC 2500 (Discovery Series) from TA instruments (New Castle, DE, USA). The samples, all 5–15 mg, were run using the following method: ramp to −50 °C, 5 min of isothermal, ramp to 200 °C, 5 min of isothermal, and cool to −50 °C. A heating/cooling rate of 10 °C /min was used for all ramps. TRIOS software (TA Instruments, New Castle, DE, USA) was used to analyze the data.

The percent crystallinity was calculated using:(1)$$X\left(\%\right)=\frac{∆H_{f}}{∆H_{f}^{o}}\times 100$$
where Δ*H_f_* is the enthalpy of fusion from the melting peak in the DSC run and $∆H_{f}^{o}$ is the theoretical enthalpy of fusion for a fully crystalline polymer. The $∆H_{f}^{o}$ value used for both PHB and PHBV was 146 J/g.

### 3.7. Size Exclusion Chromatography

PHB and PHBV samples were prepared by dissolving them (50 ± 5 °C) in HPLC grade chloroform (Fisher scientific, Waltham, MA, USA). Molecular weight analysis used an F2 flow cell with HPLC grade chloroform as the eluent. A total amount of 100 µL of each sample was injected and run through a K-805L column (Shodex, Gersthofen, BY, Germany) at a rate of 1 mL/min. The molecular weight was determined using two in-line Wyatt detectors: a Wyatt Dawn Multi-Angle static Light Scattering (MALS) detector and a Wyatt Optilab differential Refractive Index (dRI) detector (Wyatt, Santa Barbara, CA, USA). The detectors were set at 30 °C during the analysis. ASTRA software was used to analyze the MALS and dRI data. The MALS data were fit using the Zimm function as it produced the most consistent data with the smallest errors. The *dn/dc* values of 0.0256 for PHB and 0.0272 for PHBV were used for the dRI detector. Each data point is an average value of 3 measurements of 3 different samples. The value of the molecular weight of the samples was represented with standard error.

### 3.8. Density Functional Theory

DFT calculations in Figure 5 were performed using a Gaussian 16 software package (Gaussian, Wallingford, CT, USA). All the structures were geometry-optimized and frequency calculations were carried out using the B3LYP/6-31+g(d,p) level of theory. To extrapolate the degradation mechanism, three different systems—specifically, a hexamer, two trimers, and three dimers—were selected to mimic the polymer degradation. The DMol^3^ module in the Material Studio (BIOVIA Co., San Diego, CA, USA) program package was used for the DFT calculation in Appendix A [58]. The spin-restricted calculation was performed using the GGA-BLYP (generalized gradient approximation proposed by Becke, Lee, Yang, and Parr) functional [59,60] and the DNP (double numerical plus polarization) basis was set version 3.5. The convergence thresholds during the geometry-optimization procedure were set to 1 × 10^−5^ Ha for the maximum energy change, 2 × 10^−3^ Ha Å^−1^ for the maximum force, and 5 × 10^−3^ Å for the maximum displacements. The SCF (self-consistent field) tolerance was set to 1 × 10^−6^ Ha with all electron core treatment. To consider the solvent effect on the acid-catalyzed reaction, the conductor-like screening model (COSMO) [61,62] was used with a water solvent environment with a dielectric constant value of 78.54.

## 4. Conclusions

In this study, PHB and PHBV films were prepared to investigate the effects of soil humidity and chemical structure on biodegradation and physicochemical properties. The samples aged in soil and saturated with water exhibited a rapid decrease in mechanical properties, thermal stability, and molecular weight and, were fully degraded after two weeks. In stark contrast, the samples aged in soil and kept in an environment with only 40% relative humidity showed minimal changes in mechanical properties, melting temperature, degree of crystallinity, and molecular weight. Computational studies indicate that the free energy level of the protonated form of PHB is lower than that of the non-protonated form, meaning water is a very important medium for hydrolysis degradation. The FT-IR analysis, performed with a DFT calculation when PHB degrades via a hydrolysis reaction, showed a similar trend of C–O and O–H stretching peaks compared to those observed in the FT-IR spectra from the aging experiments. Contrary to the slow decrease in the crystallinity of PHBV with time, mainly due to the hydrophobic hydroxyvalerate moiety, the PHB samples under both 100% and 40% RH showed significant decreases in the degree of crystallinity within a week. In addition, a detailed degradation-time-dependent thermomechanical behavior for the two polymer chemistries was presented. Overall, this biodegradation study, as well as the understanding of the physicochemical properties of PHB and PHBV as a function of degradation time, are deemed useful for designing and optimizing the next generation of biodegradable polymers for various applications and improving sustainability and circularity in plastics.

## Figures and Tables

**Figure 1 ijms-24-07638-f001:**
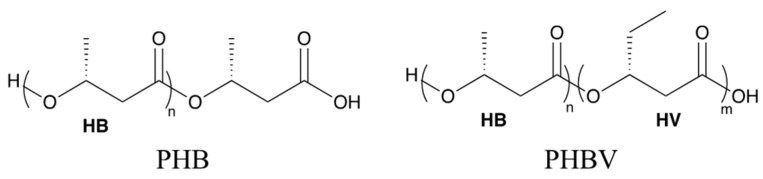
Chemical structures of polyhydroxybutyrate (PHB) and poly(hydroxybutyrate-*co*-hydroxyvalerate) (PHBV).

**Figure 2 ijms-24-07638-f002:**
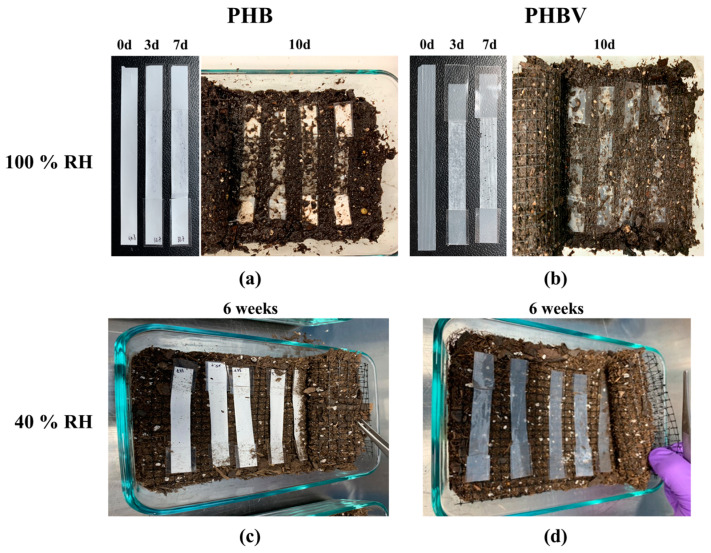
Soil degradation of (**a**) PHB and (**b**) PHBV aged in soil saturated with H_2_O (100% RH) for 0, 3, 7, and 10 days, and (**c**) PHB and (**d**) PHBV aged in soil kept at 40% RH for 6 weeks.

**Figure 3 ijms-24-07638-f003:**
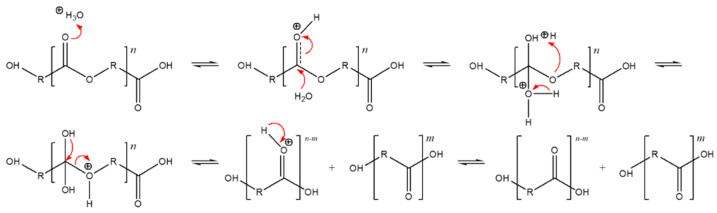
Acid-catalyzed hydrolysis reaction of polyesters, originally published in [42].

**Figure 4 ijms-24-07638-f004:**
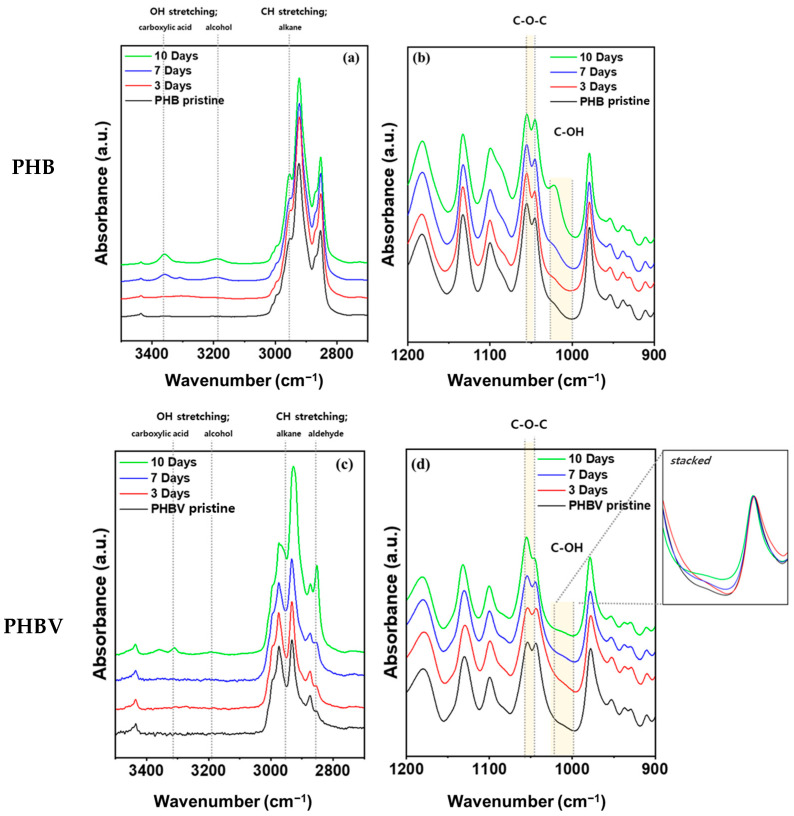
FT-IR of (**a**,**b**) PHB and (**c**,**d**) PHBV after being aged in soil fully saturated with water (100% RH).

**Figure 5 ijms-24-07638-f005:**
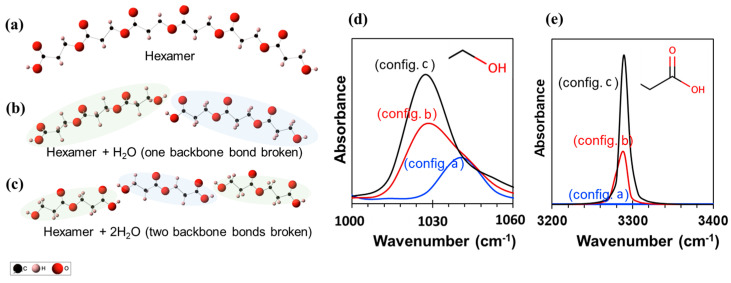
Three different chemical structures considered for the DFT calculations. (**a**) Hexamer, (**b**) two trimers with one H_2_O, and (**c**) three dimers with two H_2_O. Panels (**d**) and (**e**) show the infrared spectra for the three structures, zoomed into two different wavenumbers for the sake of clarity.

**Figure 6 ijms-24-07638-f006:**
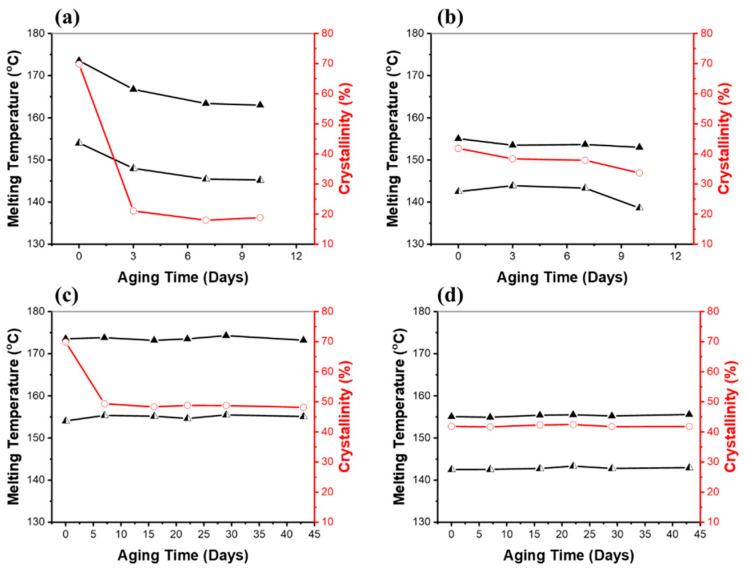
Melting temperatures (▲, ◭) and the degrees of crystallinity (○) of (**a**) PHB and (**b**) PHBV after being aged in soil fully saturated with water (100% RH), and (**c**) PHB and (**d**) PHBV after being aged in soil with 40% RH. Values were determined from DSC runs. The degree of crystallinity was calculated by dividing the enthalpy of fusion determined by DSC by 146 J/g.

**Figure 7 ijms-24-07638-f007:**
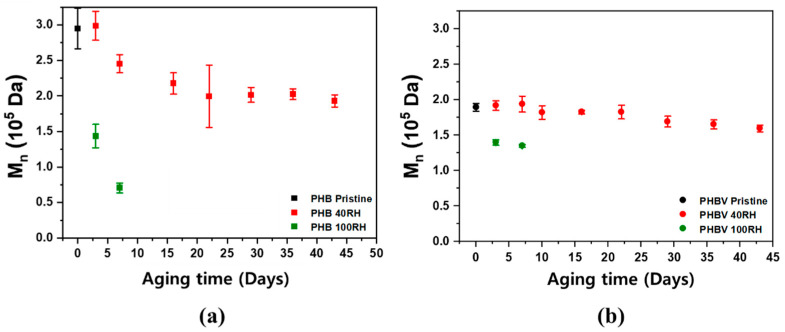
SEC determined M_n_ values for (**a**) PHB and (**b**) PHBV samples aged in soil fully saturated with water (100% RH) and at 40% RH.

**Table 2 ijms-24-07638-t002:** Mechanical properties of PHB and PHBV samples over 43 days in 100% RH and 40% RH soil conditions.

Conditions/ Aging Time		PHB		PHBV	
		Tensile Strength (MPa)	Elongation at Break (%)	Tensile Strength (MPa)	Elongation at Break (%)
100% RH	0 Day	8.43	3.73	18.75	2.59
	3 Days	3.98	0.64	9.21	0.77
	7 Days	3.84	0.52	1.18	0.70
40% RH	0 Day	8.31	3.82	18.88	2.66
	7 Day	8.46	3.79	19.17	2.83
	22 Days	8.51	3.16	19.06	2.71
	29 Days	8.24	2.71	18.96	2.84
	43 Days	7.89	2.52	19.02	2.56

**Table 3 ijms-24-07638-t003:** Summary of the changes on M_n_, T_m_, and degree of crystallinity with aging time of PHB and PHBV.

Conditions/ Aging Time		PHB				PHBV			
		M_n_ (10^5^ Da) *	Tm^1^ (°C)	Tm^2^ (°C)	Degree of Crystallinity (%)	M_n_ (10^5^ Da) *	Tm^1^ (°C)	Tm^2^ (°C)	Degree of Crystallinity (%)
100% RH	0 Day	2.95	174	154	70	1.89	155	143	42
	3 Days	1.43	168	148	21	1.40	153	144	38
	7 Days	0.71	163	145	18	1.35	154	143	38
40% RH	0 Day	2.95	174	154	70	1.89	155	143	42
	7 Days	2.45	174	155	49	1.88	155	143	42
	16 Days	2.17	173	155	48	1.85	155	143	42
	22 Days	1.99	174	155	49	1.72	156	143	42
	43 Days	1.93	173	155	48	1.59	156	143	42

* Each M_n_ data point is an average value of 3 measures.

## Data Availability

Not applicable.

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
