# Peer review of "Biodegradation Studies of Polyhydroxybutyrate and Polyhydroxybutyrate-co-Polyhydroxyvalerate Films in Soil"

_ijms, 2023, doi:10.3390/ijms24087638_

Round 1

Reviewer 1 Report

The authors did an interesting study investigating biodegradable polymers of PHB and PHBV. All information from the study was conducted precisely with the appropriate characterization. However, I would like to ask for a minor revision before accepting this manuscript.

1- The author mentions the change of polymers over time. However, images cannot be seen. Please use better images that can specify each material clearly.

2- The pH of the testing soil needs to be clarified before, during, and after a specific time to identify the effect of the polymer on the environment. The simulation of environment changes if those polymers will degrade in real life must also be mentioned.

Author Response

The authors did an interesting study investigating biodegradable polymers of PHB and PHBV. All information from the study was conducted precisely with the appropriate characterization. However, I would like to ask for a minor revision before accepting this manuscript.

(Answer) We really appreciate the time and effort that you have dedicated to providing your valuable feedback on my manuscript. We have carefully addressed all the comments. The corresponding changes and refinements made in the revised paper are summarized in our response below.

Q1] The author mentions the change of polymers over time. However, images cannot be seen. Please use better images that can specify each material clearly.

A1] We appreciate your feedback about requesting clear images of the degraded PHB and PHBV films in our article. We have added the better images of PHB and PHBV with 7 days under 100% RH and 6 weeks under 40% RH soil conditions in Supplementary Figure 1, respectively.

Q2] The pH of the testing soil needs to be clarified before, during, and after a specific time to identify the effect of the polymer on the environment. The simulation of environment changes if those polymers will degrade in real life must also be mentioned.

A2] We appreciate your feedback regarding the need for clarification of the pH of the testing soil and the simulation of environmental changes during the biodegradation experiments. Regarding the pH of the soil with aging time, we have measured them at the time of sampling on each date and found it to be still in the range of pH 6.6-6.7. It might be due to very low amount of the testing film compared to the amount of soil in each aging test batch. We apologize for not including this information in the original manuscript and have updated the text accordingly.

Regarding the simulation of environmental changes during the biodegradation experiments, we conducted the experiments in a controlled laboratory environment. Therefore, no additional simulation of the environment was performed. However, we acknowledge the importance of understanding the real-life degradation behavior of the polymers and will consider this aspect in our future research.

Reviewer 2 Report

This study on biodegradation of PHB homopolymer as well as PHBV copolymer under two different humidity conditions seems to be simple yet interesting. I suggest publishing this manuscript in Int. J. Mol. Sci. after minor revisions suggested below.

1. In Figure 3, the last two figure components have an extra -H (appear as H2O-) and they should just be -OH (alcohol).

2. In line 162, should not it be 'ester' instead of 'anhydride'?

3. In Table 3, I wonder as to why first melting point values (Tm1) are higher at 3 and 7 days compared to the pristine (0 day) PHB sample with 100% RH ?

Author Response

This study on biodegradation of PHB homopolymer as well as PHBV copolymer under two different humidity conditions seems to be simple yet interesting. I suggest publishing this manuscript in Int. J. Mol. Sci. after minor revisions suggested below.

[Answer] We're really appreciate you taking the time to share your rating with us. We are grateful to the reviewers for their insightful comments on my paper. We have highlighted the changes within the manuscript.

Q1] In Figure 3, the last two figure components have an extra -H (appear as H2O-) and they should just be -OH (alcohol).

A1] Thank you for taking the time to review our work. We appreciate your feedback regarding the typo in Figure 3, where the last two figure components appeared as H2O- instead of -OH.

We thank you for bringing this to our attention. We have corrected the typo as per your suggestion, and the revised figure has been updated in the manuscript.

Q2] In line 162, should not it be 'ester' instead of 'anhydride'?

A2] Thank you for bringing this to our attention. We have replaced 'anhydride' with 'ester'.

Q3] In Table 3, I wonder as to why first melting point values (Tm1) are higher at 3 and 7 days compared to the pristine (0 day) PHB sample with 100% RH ?

A3] We appreciate you bringing this matter to our notice. There was a mix-up with the 3rd and 7th days data points in Tm1 and Tm2. We believe they were accidentally misplaced during the table editing process. However, we have rectified the issue and clearly marked the corrected data for your reference.